# An International Comparative Reliability and Concurrent Validity Assessment of the Multi-Level Job Content Questionnaire (JCQ) 2.0

**DOI:** 10.3390/ijerph22091435

**Published:** 2025-09-15

**Authors:** Wilfred Agbenyikey, Jian Li, Sung-Il Cho, Sarven S. McLinton, Maureen Dollard, Maren Formazin, Bongkyoo Choi, Irene Houtman, Robert Karasek

**Affiliations:** 1US Department of Health and Human Services, Center for Medicare and Medicaid Services, Baltimore, MD 21244, USA; 2Department of Health Sciences, Martin Luther Health Training School, Kintampo P.O. Box 176, Ghana; 3Department of Environmental Health Sciences, Fielding School of Public Health and School of Nursing, University of California, Los Angeles, CA 90095, USA; jianli2019@ucla.edu; 4Department of Public Health Science, Institute of Health and Environment, Seoul National University, Seoul 08826, Republic of Korea; scho@snu.ac.kr; 5PSC Global Observatory, University of South Australia, Adelaide 5000, Australia; sarven.mclinton@unisa.edu.au (S.S.M.); maureen.dollard@unisa.edu.au (M.D.); 6Division “Work and Health”, Federal Institute for Occupational Safety and Health (BAuA), 10317 Berlin, Germany; 7Center for Work and Health Research, Irvine, CA 92620, USA; b.choi@uci.edu; 8Department of Medicine, University of California, Irvine, CA 92617, USA; 9TNO Netherlands Organisation for Applied Scientific Research, 2333 BE Leiden, The Netherlands; irene.houtman@tno.nl; 10Institute for Psychology, Copenhagen University, 1353 Copenhagen, Denmark; robert_karasek@uml.edu; 11Department of Work Environment, University of Massachusetts Lowell, Lowell, MA 01854, USA; 12Øresund Synergy, Frederiksberg Alle, 50 1 tv, Frederiksberg C, 1820 Copenhagen, Denmark

**Keywords:** job content questionnaire, demand-control (DC) model, ADC model, health promotion, quantitative demands, decision latitude, multi-level, pilot studies, psychometric properties, psychosocial work assessment

## Abstract

Background: This paper empirically tests the new multi-level Associationalist Demand Control (ADC) theory by applying the Job Content Questionnaire (JCQ) 2.0 that assesses both a wide range of task characteristics as well as work organizational and external-to-work psychosocial characteristics. Methods: The paper is based on four JCQ 2.0 pilot studies among 16,125 workers in Korea, China, Australia, and Germany. All pilots used the original JCQ task-level scales and then added newly developed proposed items and scales, evolving more comprehensive higher-level scales from pilot to pilot from 2005 to 2011. A brief review of the analytic process is presented, followed by an assessment of the internal consistency and concurrent validity of the final 25 multi-level JCQ 2.0 scales at the task, the organizational, and the external levels. Results: Adequate psychometric properties were established for the JCQ 2.0 pilot scales. The extended set of task-level scales was found to be robust across all samples; the new organizational scales mainly showed adequate internal consistency with α > 0.7 in Australia and Germany (tested only there) and were associated with relevant work- and health-related outcome measures as expected. Similarly, the external-to-work scales (tested only in Germany) had adequate Cronbach’s Alpha values and showed expected associations to relevant outcome scales. Conclusions: Although not all scales were available in all countries, overall, the results support the “functional similarity” of the major scale areas across the four pilot countries and support the underlying extensions of the Demand–Control theoretical constructs to the multi-level psychosocial work assessment for the promotion of workers’ health and wellbeing as suggested by the new ADC model.

## 1. Introduction

The Job Content Questionnaire (JCQ) [1] (hereafter referred to as the JCQ 1) is a well-operationalized, self-administered questionnaire designed to measure psychosocial work characteristics of jobs, focusing mainly on the task/job level. It has been found to have both satisfactory reliability and validity in assessing its underlying theoretical constructs. Moreover, it predicts both negative (e.g., stress-related illness) and positive (e.g., job engagement, satisfaction) work-related outcomes across occupations, cultures, and countries [2,3,4,5,6,7,8,9,10,11].

However, profound changes in the way work is organized and carried out around the world have taken place over the last decades due to globalization and strong free-market-based socioeconomic policy. E.g., the service sector has grown substantially while, at the same time, the world of work has been faced with both the opportunities and threats brought about by digitalization. The need to revise the JCQ 1 has become evident in order to effectively measure and assess the overall social and psychological work experience in today’s world of work; for details, see Karasek et al. [12].

The JCQ 1 has been further developed into the JCQ 2.0. The latter provides a holistic understanding of stressors of the contemporary work environment. Such an understanding is crucial in order to develop interventions that aim at redesigning work to improve occupational health. The domains of the JCQ 2.0 capture the drastic changes in the way work has been organized in the past decades largely due to economic globalization and neo-capitalist policies. The JCQ 2.0 significantly extends the JCQ 1 while maintaining generalized versions of the constructs of Demand–Control (DC) theory [13,14], on which the JCQ 1 is based. The additional scales added to upgrade the JCQ 1 are based on the Associationalist Demand Control (ADC) model, which is described in detail in this Special Issue by Karasek et al. [12]. Therefore, the JCQ 2.0 includes both the task/job level psychosocial factors of the work environment as well as additional areas such as the context of the job at the local employment site (the “organization level”) and external labor market, as well as work–family challenges. The JCQ 2.0 captures the risk factors, root causes and stressors that emanate from the modern organization of work that may have been missed by the JCQ 1.0. Employee health promotion and stress preventive initiatives can be more effective if the entire set of risk factors is known.

The aim of this paper is to use psychometric analyses to establish items and scales for a new JCQ 2.0 psychosocial workplace risk assessment tool that can function in an equivalent manner across different cultures and populations. The paper assesses the internal consistency and concurrent validity of the JCQ 2.0 items and scales using data from Korea, China, Australia, and Germany—four countries where studies have suggested the presence of workplace stress [15,16,17,18]. The paper reflects an ambitious goal: with work organization varying across different socio-geographic areas—regions, nations, cultures, companies—a cross-culturally valid assessment tool for psychosocial workplace characteristics would undoubtedly aid in assessing these expanding risks in an internationally comparative context.

### 1.1. The Developmental Process

During the development of the JCQ 2.0 in the four countries, items and scales evolved considerably across a span of six years: from Korea in 2005, China in 2006, and Australia in 2009 to Germany in 2011. Therefore, some of the organizational and external scales are only available in the later Australian and German pilots. The JCQ 2.0 development process is graphically presented in Figure 1. It encompassed three stages and started with multiple international workshop meetings with a core of committed researchers that—in Stage 1—compiled all critique regarding the JCQ 1. As a consequence of these considerations, a ‘rough design’ of the new and extended JCQ was formulated in Phase 2. The JCQ 2.0 scales evolved on the basis of two generations of theory: the first set of pilots in Korea and China was based on a rough theory (Figure 1: Step 2A) with expansions based on both creative workshop discussions and the JCQ 2.0 rationale document (available at www.jcqcenter.org, accessed on 3 July 2019). At that stage, only an outline for the need for organization-level Demand, Control, and Support concepts was available. However, there was yet not sufficient detail to generate a fully integrated set of new concepts, scales, and items beyond the task level. Korea provided the first opportunity to test the broader, if basic, JCQ 2.0 scale set with a large urban transit sample (Sung-Il Cho). In Western China, the first representative urban population survey was undertaken (Jian Li).

The second stage of the pilots in Australia and Germany (Figure 1: Step 2B) came after the development of the new more coherent and rigorous ADC theory base [12,19]. These pilots incorporated a much broader set of organizational scales in Australia, accompanying Maureen Dollard’s own organizational scale development [20]. The final JCQ 2.0 version in the German pilot (Maren Formazin) contained many new organizational and external scales and was the version that most fully assessed the expanded and generalized system-theoretic version of the DC model: the ADC theory as presented in Karasek et al. [12].

The current global economic and organizational contexts necessitate a multi-level theoretical framework to understand the complexities of the psychosocial work environment. Building on the DC model that underpinned the JCQ 1.0, ADC theory expands the social relational aspects of working conditions. It has significant implications for addressing global occupational health challenges, particularly in relation to mental health, work-related stress, and sustainable work environments. By understanding the social relational aspects of working conditions, organizations can develop strategies to promote employee wellbeing, reduce work-related stress, and foster environmentally sustainable practices. The JCQ 2.0’s new conducive behavior scales can underpin a social development process that contributes to a more sustainable and healthy work environment. This framework can inform organizational practices, leadership development, and workplace interventions aimed at promoting employee wellbeing.

### 1.2. Hypothesized Scales of the JCQ 2.0 Used in the Four Pilots

In spite of the new workplace challenges, recent international literature reviews have confirmed the utility of the often-utilized Demand, Control, Support (DCS) conceptual structure [21,22]. ADC theory [12], which is the basis for the JCQ 2.0, is an updated version of the classic Demand–Control (DC) model that attempts to extend the classic narrative to support assessment of psychosocial work environment conditions in the context of our current complex global economy—with the necessary multiple levels of scales and cross-level effect hypotheses—now based on multi-level versions of the Demand, Control, and (re-labelled) Stability–Support constructs.

Possibilities of such generalization are found in a physiological extension of (a) the classic DC “job strain” hypotheses via Stress-Disequilibrium Theory (SDT) [23] and (b) a further extension for the “active work” hypothesis to the organization and economy level via Conducivity Theory [12,23,24]. In an integrated form, these represent the broader logical platform labelled ADC theory [12].

In the context of the more generalized ADC hypotheses, we expand the previously adopted concept of “social support” and re-label it, for multi-level usage, as Stability/Support [12]. The new concept provides both a proactive as well as a protective component, it better accommodates the dynamism of modern work life, and it supports an expanded set of organizational-level scales. It can address new work demand challenges that require multiple levels of control at work and skill possibilities—hypothesized to function in an integrated manner within and across levels (workers, departments, etc.). The JCQ 2.0 organizational scales represent, in most cases, an assessment of the full workplace as seen by the worker below. The ADC theory’s multi-level concept reflects the need to allow the worker the possibility of maintaining an easy-to-achieve personal equilibrium in daily working life—and, at the same time, reflects the organizations’ need to continually bring in new resources for effective environmental adaptation and further growth.

ADC retains the concept of three underlying factors of Demand, Control, and Stability–Support. All scales belonging to the JCQ 2.0 are hypothesized to be indicators of one of these factors. Note that this aspect will not be covered in this paper but is empirically tested by Formazin et al. [25].

#### 1.2.1. JCQ 2.0 Hypothesized Demand Scales

In the theory context of JCQ 2.0, it is the individual’s job that requests resources relating to the employee’s skill application area and demands that require “output” to be delivered [12].

There are three JCQ 2.0 demand scales proposed at the task level as opposed to two in the JCQ 1: quantitative demands, emotional demands, and physical demands. (1) The psychological demands scale as it was in the original JCQ 1 was re-labelled quantitative demands for the JCQ 2.0 to avoid ambiguity. (2) A new emotional demands scale refers to the requirement for workers to regulate their feelings and expressions in order to achieve organizational goals when dealing with other people, i.e., customers or clients [26]. (3) The physical demands scale assesses physical demands and hazards outside the primary theory area, but is retained from the JCQ 1 because of its significance for workload in many occupations and countries.

Two additional JCQ 2.0 scales are needed to assess demands at the organizational level. (1) The organizational restructuring scale assesses adaptive changes in the organizational structure, arising either because of company instability or growth, both processes being stressful and requesting adaptation [27,28,29,30]. (2) The organizational disorder scale assesses the degree to which organizational instability relating to ineffective and chaotic operational or management processes causes its own set of extra job demands for employees [31].

#### 1.2.2. JCQ 2.0 Hypothesized Control Scales

The JCQ 2.0 has three scales to assess control at the task level: decision authority, skill discretion, and conducive development. The skill discretion scale is modified from the JCQ 1.0 with the elimination of some items while still referring to skill usage, being creative, and developing oneself in the job. The short decision authority scale, referring to the worker’s ability to control his or her own activities on the job, remains unchanged from the JCQ 1.0. The conducive development scale, focusing on the organizational facilitation of employee ability development, has been newly added to update the related skill development scale in line with the current global economic context and the concept of collaborative self-actualization [12,19,24,32]

Three JCQ 2.0 control scales are proposed at the organizational level. Two reflect relatively routine communication—assessing whether the employee has organizational decision latitude and procedural justice. The organizational decision latitude scale [33] assesses whether the employees have significant possibilities of contributing their own perspectives or are consulted in matters that affect them. The procedural justice scale [34,35] assesses whether decision-making procedures are fair (i.e., with input from affected parties and the avoidance of suppressing bias). The third JCQ 2.0 control scale is the conducive communication scale, which captures aspects of innovation brought about by co-partnering between employees and customers and by a focus on customer needs for output [19,24,32].

#### 1.2.3. JCQ 2.0 Hypothesized Support–Stability Scales

The task-level social support scales from the JCQ 1.0 retain their classic meaning in the JCQ 2.0, reflecting individual-level job strain and stress coping through tight and explicit interpersonal linkages. However, in the context of the more generalized ADC hypotheses, we expand the previously adopted concept of “social support” to include both proactive as well as protective components To better accommodate the new organizational level stability and support-related scales in the multi-level context, we re-label them as Stability–Support [12].

To assess JCQ 2.0 task-level social support in a manner that maintains both the instrumental as well as the affective aspects [14] of the JCQ 1 task scales, the question set is retained but modified for supervisor support and coworker support. An expansion of each scale adds a highly scale-correlated question relating to respect from the supervisor/coworkers, reflecting a close relation with Siegrist’s ERI model [36]. “Coworker” is conceptualized as “people one works with”, giving it broader coverage than just colleagues (e.g., including suppliers or business partners). Two additional support scales are added at the task level: collective control refers to collective forms of control among colleagues through the development of work groups [37,38,39]. The negative acts scale is a short, summary-level measure of the adverse aspects of social relations at work including social isolation and harassment [40,41,42].

JCQ 2.0 Stability–Support on the organizational level is assessed by three scales. Psychosocial safety climate (PSC) is a specific dimension of organizational climate referring to shared perceptions regarding “policies, practices, and procedures for the protection of worker psychological health and safety” [43,44]. The organizational rewards scale assesses recognition and financial rewards based on Siegrist’s ERI model [36]. A new consideration of workers’ interests scale extends the reward concept to the importance of employees’ wellbeing in the face of organizational changes. The latter two scales, i.e., organizational rewards and consideration of workers’ interest, can be combined to form the scale of organizational fairness, which focuses on a fair division of the costs and benefits of organizational participation and transformation. The implications of the combined set of JCQ 2 organization-level scales for the protection against health risks and the promotion of health and wellbeing is further discussed by Karasek et al. [12].

#### 1.2.4. JCQ 2.0 External-to-Work Scales

The multi-level ADC theory can be extrapolated to the level that is external to the organization itself, reflecting new levels of external labor market risk [12] and having an impact on the organizational and, in turn, the task level. The JCQ 2.0 attempts to measure the psychosocial aspects of “external-to-work”, starting with existing indicators and expanding them to encompass a larger range of “outside-work” structures, including effects of the global economy

Work–family conflicts are assessed since they comprise a major part of the contemporary work-life and moderate the impact of direct job-related risks.

Four scales covering job insecurity in a socially-expansive manner are introduced: (a) an “overall” scale (job insecurity—personal), (b) a “long-term” job insecurity career-related scale, (c) a scale reflecting a worker’s use of social resources from other life spheres (job insecurity social relations) to secure a job, and (d) a preliminary job insecurity—institutional support scale assessing public support for job search (not here tested). Additional effects of higher-level demands of the global economy—demands and insecurity scale and the workers’ perception of labor market “control” in his/her labor market position utilize questions that have been successfully tested in the JCQ 1.7 [45,46].

The external-to-work scales are supposed to be applicable in diverse socioeconomic and cultural contexts to address public health disparities. Gaining insight into these aspects can inform tailored interventions and policy decisions aimed at reducing health inequities and promoting healthy work environments. The scales are meant to be used to study the impact of external factors on worker wellbeing also in low- and middle-income countries while it is acknowledged that they might need to be adapted to capture unique cultural and socioeconomic factors affecting worker wellbeing in diverse populations. By assessing the external factors and worker wellbeing, researchers and practitioners can gain a deeper understanding of their interplay that will, in turn, help them develop targeted strategies to address public health disparities and promote healthy work environments across different socioeconomic and cultural contexts, reducing health inequities in the long run.

The following Table 1 presents an overview of the JCQ 2.0 recommended scales at three levels: task, organization, and external-to-work. It indicates whether the scales were based on the old JCQ 1 scales, were derived from the literature, or were newly devised scales.

### 1.3. Hypothesized Relations of JCQ 2.0 Scales to Outcome Variables

Associations between JCQ 2.0 psychosocial scales and outcomes follow the general hypothesis structure of the original DC model extended to multiple levels. The expanded list of JCQ 2.0 scales—now 25—brings the advantage of a much more detailed assessment of risks. However, the empirical testing of the JCQ 2.0 necessitates a “simplification” of the vast multitude of potential empirical associations that could be tested for the ambitious goal we pursue: We wish to validate the JCQ 2.0

(1)by establishing scales of adequate internal consistency;(2)with scale intercorrelations as derived from theory, i.e., higher correlations among scales belonging to the same underlying construct of demand, control, or stability–support;(3)with associations with relevant work-related outcome measures as hypothesized by ADC theory.

Importantly, the validation shall not be restricted to one country but will encompass four countries, striving for the first international validation of the JCQ 2.0.

To simplify the following presentation of the concurrent validity results, umbrella “positive” and “negative” labels for dependent variables will be utilized for the interpretation of the tests. Furthermore, scale orientations of the independent variables will be either maintained or reversed in such a way that will allow for a simplified confirmation of the classic DC hypotheses directionalities.

The DC model’s active work hypotheses will be considered confirmed when they are associated positively with the “positive work-related outcomes”. Similarly, the DC model’s job strain hypotheses will be considered confirmed when they associate positively with the “negative illness outcomes”.

Independent variable scales will maintain their nominal directionality where higher scores are hypothesized to predict high levels of wellbeing, i.e., “positive outcomes” (e.g., the decision authority scale). Alternatively, the scale directions will be reversed for “negative outcomes” so that a lower scale score reflects stronger risk. Thus, in tests with decision authority—where low control is presumed to be a risk factor for chronic disease [23]—the independent variable’s association-coefficient sign will reflect higher scores at “lower control” in the case of hypothesis confirmation.

There will be no such reversal when the nominal scale meaning is typically associated with risk (i.e., the quantitative demands scale) and the outcomes predicted are “negative” (illness) outcomes.

## 2. Materials and Methods

### 2.1. Study Population

The total study population comprised 16,125 working men and women across a range of occupational settings from four cross-sectional pilot studies in Korea, China, Australia, and Germany. Table 2 presents demographic characteristics for the four samples.

The Korean pilot was conducted with 7407 transit workers in Seoul, South Korea in 2005 (response rate was 74.8%). The Chinese pilot was a community-based random sample survey of 2178 employees from Kunming, China in 2006 (response rate 87%). The Australian pilot was a random population based Australian Workplace Barometer survey of 4214 employees between 2009 and 2011 in four Australian States (New South Wales, Western Australia, South Australia, and Tasmania) and two Territories (Northern Territory and Australian Capital Territory) through computer-assisted telephone interviews, with a response rate of 25%. The German pilot was a mail out, population-based random survey of 2326 employees (civil servants, white-collar workers, and blue-collar workers; freelancers and self-employed were excluded from analyses) in the Ruhr area of Germany in 2011 (23.8% response).

### 2.2. JCQ 2.0 Item Development, Translation, and Scale Score Estimation

As noted in Section 1.2, the JCQ 2.0 retained the original core JCQ 1 scales and items assessing task level quantitative demands, skill discretion, decision authority, supervisor and coworker support, physical demands, and aspects of job insecurity to ensure comparability with the original instrument. However, all items were critically assessed based on available empirical data, leading to the decision to refrain from using some items in the JCQ 2.0.

With regard to the quantitative demands scale, the item “work hard” has been shown to be ambiguous as it not only refers to psychological demands but also to physical demands and has differential associations with outcome measures for specific work groups, e.g., when comparing white- and blue-collar workers [47]. Also, the question is difficult to translate consistently across languages. Even though the question has the advantage of using language more broadly familiar to a sample that includes participants with lower education, the item was not kept in order to render the scale more clear.

A second item, “repetitive work” from the skill discretion scale, has been shown to be psychometrically problematic because of low scale-item-correlations [3,48,49,50,51,52,53] and, therefore, was also excluded.

However, both items are suggested to be retained in a special “archival” JCQ 2.0 version for users with need for historical comparison. E.g., “repetitive work” as hypothesized positively correlates with negative outcomes such as poor self-rated health, depression, and burnout and is also negatively related to the positive outcome of job satisfaction across different cultures. Primarily, “repetitive work” measures specialization of labor within the job—a core feature of the Western, mass-production, industrialized work process and economic logic already identified by Adam Smith in 1776 [54] as the productivity foundation for a free-market, physical-production-based economy. Currently, the distribution of repetitive work remains important, now representing re-allocation of labor content across economies. Its relative disappearance in advanced economies has contributed to political dissatisfaction via job loss, robotization, and limited work re-qualification [55,56].

In addition to the scales from the JCQ 1, the new literature-based scales included in the JCQ 2.0 were taken from available instruments, e.g., with regard to PSC [43,44] and organizational rewards [36]. Scales for which no suitable instruments were available were newly developed based on theoretical considerations, e.g., with regard to organizational chaos.

Following Choi et al. [57], a thorough translation process was applied to translate the questionnaire—originally developed in English—into three languages, striving to minimize different meanings of items across countries. The translated questionnaires were all back translated and approved by the JCQ Center.

Consequently, across the pilots, items in the scales were very similar in content but not always exactly identical in wording. Some pilots tested more or fewer items in a particular domain or scale.

Most items used a four-point Likert scale from 1 (*strongly disagree*) to 4 (*strongly agree*). For the correlation of scales, the JCQ 1 scales were calculated based on the recommended formulas [58]. For the comparative concurrent validity of the scales, simple mean (at zero) question score additions were used in the computation of the scales.

### 2.3. Measures of Outcome Variables

The JCQ 2.0 pilot design focused on workplace assessment and thus did not initially specify dependent measures for the two early pilots, i.e., in Korea and China, the JCQ 2.0 pilot studies often assessed wellbeing only utilizing single question indicators, not allowing for meaningful conclusions regarding positive outcomes in these two countries. Hence, only negative outcomes will be considered for both China and Kore.

For the two later pilots in Australia and Germany, a comprehensive set of six dependent variables was assessed for three positive outcomes, namely job satisfaction, job engagement/commitment, and intention to stay/intention to leave [59,60,61,62,63,64,65], and three negative outcomes, namely burnout, depression, and poor self-rated health [66,67,68,69].

### 2.4. Data Analysis

To develop the JCQ 2.0, a multi-stage process was applied, first based on separate analyses of the four a priori pilot studies and then utilizing an “integrated analysis” across all pilots for the finally selected items in the recommended final scales. All analyses were conducted separately for each country in order to ensure that possible differences in item formulation would not distort results. Moreover, as has been described above, not all scales and items were available in all countries, rendering a fully combined analysis impossible.

#### 2.4.1. Handling Missing Data

For each survey item, there were at least some missing responses per country, varying between 0.2 and 5.6%. Although non-response was not uniformly low for any particular category of questions, confining analyses only to the complete cases for all subjects on all items would have led to disregarding 15 to 32% of the respondents per pilot. Data missing completely at random (MCAR), implying that causes of missingness were unrelated to the data [70], was indirectly tested in the samples by ascertaining which socio-demographic variables were predictive of missingness using logistic regression in SPSS 17.0. Results imply that data was not MCAR. Thus, to correct for possible bias, all data sets were re-created separately 10 times using SPSS’s “fully conditional specification (FCS)” procedures. These multiple imputation data sets were used for correlation analyses and assessment of internal consistency. Concurrent validity analyses utilized SAS-v9.3 Proc Logistic regression to test each scale separately, using listwise deletion for missing items within each scale and adjusting for age.

#### 2.4.2. From the a Priori Pilot Scales to the Final JCQ 2.0 Scales

The overall process of validating the JCQ 2.0 questionnaire is shown in Figure 2. The “a priori” initial JCQ 2.0 question sets were developed from theory, and a stepwise process involving multiple iterations of exploratory factor analyses, assessment of internal consistency, and associations with relevant outcome measures was applied (Figure 2, Phase I). We here note that exploratory factor analyses supported the hypothesized DCS-S structure of the JCQ 2.0 items across all pilots, providing support for the further integration noted in the ‘Results’ section. Details of the a priori pilot analyses are beyond the scope of this paper but are available on request at the JCQ Center (www.jcqcenter.org, accessed on 3 July 2019).

Following factor analyses, five criteria were applied to select the “candidate” JCQ 2.0 items from within the set of four pilot studies and to select the final recommended scales for the final JCQ 2.0 for assessment in the analyses below (Figure 2: Phase IIA):(1)the theoretically-anchored scale descriptions,(2)a scale-related set of items common across all pilots (where applicable),(3)assessment of internal consistency, first based on a priori scales,(4)item-by-item concurrent validity analyses, first assessed in the a priori scales,(5)attempt to assess each scale with a small number of items (no more than three if possible) with strong cross-sectional associations to maximize usage efficiency of the final questionnaire since the scales cover a broad range of differentially important content areas, useful to measure separately.

This procedure left few items in each scale, which was a necessary efficiency given our coverage goals. However, it also reduced Cronbach’s Alpha, especially for the early-stage pilots, since only common-to-all-pilot items were utilized, giving priority to the later-stage items. Consequently, this decision meant that international comparability and coverage breadth research goals had effectively given priority to testing similarity of scale content over testing internal scale consistency. Item differences between countries were usually not large, and very often non-existent. However, in some cases, between-country differences did raise questions, as will be noted.

To finally define and refine the Recommended Researcher JCQ 2.0 scales, an additional stage of psychometric testing was applied to candidate scales evolved from the pilot processes (Figure 2, Phase IIB), as will be described in the next section.

#### 2.4.3. Scale Quality Assessments

For indicators of scale quality, we utilized Cronbach’s Alphas: considered poor when lower than 0.50, acceptable when up to 0.70, and good when higher [71].

For concurrent validity testing, the associations between JCQ 2.0 items (for the a priori original version) and scales and dependent variables were assessed by logistic regression, adjusted for age, and stratified by sex, yielding standardized odds ratio (SOR) to facilitate comparison between independent variables [72]. The SOR measures the change in the odds of each dependent variable (the six outcome measures) for each standard deviation change in the independent variable (the JCQ 2.0 scales). This requires the outcome measures to be dichotomized. For our analyses, roughly the uppermost 20% of each outcome were defined as “cases” to avoid the misclassification that the middle ranges of the scales might bring. SORs yield a standard assessment of relative association strength of different independent variable items—within a proposed scale area—with the same outcome content area [12,73] across survey populations. We identified a SOR of 1.55 to indicate a substantial strength of association. In broad population samples, a SOR of 1.55 or higher implied a dependent variable variation of about 2.4:1 from the top to the bottom tertile of the independent variable. It has to be pointed out that this was not the same as “significance level”.

The addition of SOR testing for the items helped complement considerations of internal consistency with considerations of concurrent validity. Reviewing internal consistency together with concurrent validity based on the SOR’s strength of association criterion allowed a process with selection criteria capable of assessing common “cause” association strength at a detailed question level across pilots. This was associated with items assessing common effects even when not exactly similar in wording.

## 3. Results

### 3.1. Recommended JCQ 2.0 Scales

The new recommended JCQ 2.0 scales covered aspects of the psychosocial work environment at three levels:(1)ten scales at the task level: five revised classic JCQ 1 scales, one literature-based scale, and four newly defined scales;(2)eight scales at the organizational level: three from the literature and five constructed from new theory;(3)seven scales at the external-to-work level: one revised from the JCQ 1, one from the literature, and five newly constructed from theory (one not tested).

Table A1 in Appendix A presents an overview of the JCQ 2.0 recommended scales and keywords for the items included in the scales.

### 3.2. Internal Consistency of the Recommended JCQ 2.0 Scales

Table 3 reports on the Cronbach’s Alpha coefficients of the JCQ 2.0 recommended scales. Out of the 57 separate scale tests across four pilots, 81% were based on three or fewer items (with 30% of those with only two items). For the German pilot, Cronbach’s Alpha ranged from 0.57 to 0.90, with 73% of the 24 scales with an alpha above 0.70, indicating good internal consistency. For the Australian pilot scales, Cronbach’s Alpha ranged from 0.50 to 0.86; with 62% of the scales above 0.70, indicating good internal consistency.

The Korean and Chinese pilots had mainly moderate and relatively lower Cronbach’s Alpha in Table 3. This was due to the fact that the Cronbach’s Alphas were based on the relatively small number of relevant items common to all pilots. In addition, the German and Australian pilots both had a theory-anchored head start and the longest instruments (see Figure 2). Over 80% of the computations had three or fewer items (44% of those with just two items) that were common across all final recommended JCQ 2.0 scales. This led to several less than normally acceptable Cronbach’s Alphas for the scales in Korea and China.

### 3.3. Intercorrelation of the Recommended JCQ 2.0 Scales

Table 4, Table 5, Table 6 and Table 7 show the inter-scale correlation coefficients for the new JCQ 2.0 scales for Korea, China, Australia, and Germany, respectively. The basic similarity of scale associational patterns across pilots is generally confirmed. Note that all of the inter-scale correlations in this section are further investigated using confirmatory factor analysis in a related paper in this Special Issue by Formazin et al. [25], which presents the interrelations in relation to demand, control, and stability–support as underlying constructs.

Demand and control scales empirically remained independent constructs in all four JCQ 2.0 pilots with a correlation of quantitative demands and decision latitude ranging between *r* = 0.04 and *r* = 0.21 (Table 4, Table 5, Table 6 and Table 7). This result confirms the existence of a job strain group in the new JCQ 2.0 in line with its original conceptualization from the DC model [13,14].

Skill discretion and decision authority remained moderately correlated but were confirmed as independent scales in all four pilots (Table 4, Table 5, Table 6 and Table 7). Their theoretical linkage, as the often-used decision latitude construct [13,14], has been much discussed. The conducive development scale—which was only tested in Germany—demonstrated strong associations with both skill discretion and decision authority, with *r* = 0.52 and *r* = 0.60, respectively (Table 7).

In all four pilots, quantitative demands had low correlations with physical demands (0.03 ≤ *r* ≤ 0.23; Table 4, Table 5, Table 6 and Table 7), confirming independence, and low-to-moderate correlations (0.06 ≤ *r* ≤ 0.46; Table 4, Table 5, Table 6 and Table 7) with emotional demands, also confirming independence, but in this case with more closely related scales. The merits of adding emotional demands and their non-redundancy in relation to quantitative demands are further shown in Formazin et al. [74] in this Special Issue.

Supervisor support and coworker support correlated moderately, supporting both a broader social support scale and also two separate scales (*r* ranged between 0.39 and 0.47; Table 4, Table 5, Table 6 and Table 7).

Only tested in the German pilot, the collective control scale was significantly and moderately to strongly correlated with supervisor support (*r* = 0.45; Table 7) and coworker support (*r* = 0.62; Table 7). Although theoretically differentiated and predictively successful [38], this scale’s independence should be further demonstrated. Negative acts, in the expanded task support area, correlated moderately and negatively with all social support scales: *r* = −0.38 for supervisor support, −0.44 for coworker support, and −0.46 for collective control (Table 7).

Overall, task-level control and social support scales correlated more strongly with each other than with the demand scales. The former was line both with ADC theory [12] as well as with conceptualizations of the Job Demands–Resources model (J-DR-model) [75,76,77,78]. Typically, scales involving quantitative demands have inter-scale correlations that are less consistent than those of other tested JCQ scales [1], either because of a scale’s lower objective validity as assessed by inter-occupational variance [14] or because of potentially large cultural variation.

To summarize, in line with theoretical expectations, the enhancement of the established JCQ 1 with new scales on the task level led to a wider assessment of the work environment with no redundancy.

At the organizational level, there was a similar pattern with stronger inter-scale correlations between control and social support scales as compared to demands. In both Australia and Germany, where organizational-level scales were available, organization-level demands were consistently negatively correlated with both the organizational-level control and the organizational-level stability–support scales: a finding that replicated typical associations between those scale sets at the task level. Generally, the organizational control and support scales correlated moderately to strongly with each other, supporting the conceptualizations of ADC theory [12].

In line with theoretical expectations, specific organization-level scales were closely correlated to specific task-level scales in the same content area, a topic that is further explored by Formazin et al. [25]. Organizational decision latitude remained mainly moderately correlated with the components of task-level decision latitude across all four pilots (Table 4, Table 5, Table 6 and Table 7), confirming the a priori theory of different but related constructs. Conducive development and conducive communication were strongly interrelated (*r* = 0.58) in Germany (the only country with both scales available; Table 7).

The task-level social support scales also correlated moderately to strongly with the social support and control organizational-level scales in China, Australia, and Germany (Table 5, Table 6 and Table 7).

The external-to-work scales that were measured in Australia (work–family conflict) and Germany (where there were also three job insecurity scales, global economy—demand and insecurity, and labor market control) in general showed low correlations to other concepts (mostly *r* < 0.30; Table 7).

What was striking were the comparably high correlations of work–family conflict with quantitative and emotional demands on the task level (both *r* = 0.44) in Australia (Table 6) and, similarly, in Germany (*r* = 0.41 and *r* = 0.29, respectively; Table 7). In addition, work–family conflict correlated strongly with both organizational restructuring (*r* = 0.61) as well as job insecurity—career (*r* = 0.89) in the German data (Table 7). This implies that work–family conflict is especially high when workers are faced with high demands at work.

Moreover, in the German data, job insecurity—career showed strong negative correlations with both conducive development (*r* = −0.60) and the organizational-level control scales (−0.45 ≤ *r* ≤ −0.51; Table 7), indicating that when job insecurity is high, workers are granted less decision latitude and less room for skill development.

### 3.4. Correlation of the Recommended JCQ 2.0 Scales with Relevant Outcome Measures

The results of logistic regression between the JCQ 2.0 scales and three negative health outcomes—namely burnout, depression, and poor self-rated health—for all four pilots (Korea, China, Australia, and Germany) separately are shown in Table 8. For the Australian and the German pilots, three positive outcomes (job satisfaction, work engagement/affective commitment, and low intention to quit/high intention to stay) were additionally tested for their independent associations with all the scales; the results are presented in Table 9.

Higher SORs indicate stronger associations, with associations greater than 1.0 implying consistency with the DCS-S hypotheses’ predicted directionality. Altogether, more than half of the SORs in Table 8 and Table 9 were 1.55 or higher in Australia and Germany, indicating substantial associations, and with association directionality as predicted in classic DCS-S hypotheses.

Table 8 shows that health problems were mainly positively related to demands and negatively related to both control and its differential aspects as well as indicators of stability–support. These findings were the same for the psychosocial risks at both the task and the organizational levels. Across the pilots, associations were typically stronger with burnout, followed by depression, and were least strong for poor self-rated health. Only for Germany, burnout and depression had similar strength. The overall associations were substantially weaker in China, and especially in Korea.

Table 9 shows the classic DCS-S relationships with positive outcomes: positive associations with the JCQ 2.0 control and stability–support scales and negative associations for quantitative demands and emotional demands (higher numbers here indicating stronger consistency with hypothesized negative outcomes). The associations were strongest for job satisfaction—an outcome measure most explicitly related to work.

Thus, despite the imprecisions involved in cross-sectional data, almost all scales significantly explain variance in both positive and negative outcomes (a) in a consistent manner, (b) across countries, and (c) as hypothesized. Further examinations of the above dependent variable associations are discussed for “composite” DCS-S scales by Formazin et al. [79], and in detail, scale-by-scale, in a related publication in this Special Issue [74].

## 4. Discussion

The JCQ 2.0 was developed to measure psychosocial risks at the level of the task, the organization, and external to work. This was in line with the International Labour Organization’s (ILO) core value statement, which describes a multi-level conception of work organization comprising the external context, the organizational context, and the work task context [80].

The present study was undertaken to test the psychometric properties of the JCQ 2.0. It provided exploratory psychometric information on all items and scales that were tested across four countries, namely Korea, China, Australia, and Germany. Overall, the findings show adequate support for the psychometric properties of the JCQ 2.0 scales in terms of internal consistency, inter-correlations of scales, associations with relevant outcome measures, and functional similarity across countries.

During the process of JCQ 2.0 pilot testing (Figure 1 and Figure 2), the questionnaire evolved, leading to both slightly different items across pilots as well as a sequentially expanding set of scales. However, the full set of JCQ 1 core scale items was retained in all pilots to retain consistency with a vast body of JCQ-based research, which remains relevant even in spite of aging [81].

Regarding internal consistency, the results were mainly satisfying, with Cronbach’s Alpha being in many cases higher than 0.70, implying sufficient internal consistency. It has to be noted that this was especially the case in the two later pilots in Australia and Germany, where the majority of scales had three items, while this was often not the case in the two earlier pilots in Korea and China. The results underline the importance of scales that comprise at least three items. With the final recommended JCQ 2.0 scales, this aim could mainly be achieved and further efforts should make sure that all scales are assessed with at least three items.

For the final recommended JCQ 2.0 scales, scale intercorrelations were mainly as anticipated:(1)between groups of task-level scales,(2)between organizational level sets of scales (control-related, demands-related, and stability–support-related),(3)across levels,(4)across pilots.

This was further empirically demonstrated by Formazin et al. [25], especially in the Australian and German pilots.

These associations were theoretically consistent with the anticipated relationships and consistent with the DC model in its generalized multi-level extension as ADC theory [12]:

Skill discretion and decision authority were shown to be related yet distinguishable concepts, implying the possibility of both considering them in a combined way via decision latitude as well as, similarly, their merit in further exploring their individual aspects as is shown by Formazin et al. [74]. In addition, conducive development could be shown to be a complementary concept to the latter two.

Similarly, both quantitative and emotional demands were shown to be substantially correlated yet distinguishable, implying differential associations with outcome measures, as is shown by Formazin et al. [74].

Regarding social support, collective control and negative acts emerged as related yet distinct concepts in relation to supervisor and coworker support, respectively.

Analyses replicated that control and support are positively related concepts and that they correlated more strongly with each other than with demands on both the task and the organizational levels. Moreover, control and support consistently had negative associations with demands, a result in line with both ADC theory [12] as well as the J-DR-model [75,76,77,78].

With the consideration of the newly developed external-to-work scales in the JCQ 2.0, it was possible to investigate how the latter are related to scales on the task and the organizational levels. Two interesting aspects emerged from these analyses. On the one hand, work–family conflict showed strong associations with demands on both the task level as well as the organizational level, implying that in order to minimize a negative impact on the workers’ lives outside work, the workers’ demands need to be designed in a manageable way. On the other hand, workers experienced low organizational control when they perceived a high level of job insecurity regarding their career, indicating a need for suitable interventions.

In relation to concurrent validity, there was consistency in the direction and strength of the scales’ relations to positive and negative wellbeing outcome indicators in all four countries, strengthening implications regarding concurrent validity for the new scales. Associations were highest for those outcome measures that were very clearly related to work, i.e., burnout and job satisfaction, and comparably lower for outcome measures that were influenced by a lot of outside-work-factors, e.g., self-rated health.

Associations with outcome measures were comparably lower in the earlier two pilots in China and Korea compared to the later two pilots in Australia and Germany. Here, one has to bear in mind that in the later pilots, both the JCQ 2.0 scales as well as the outcome measures were assessed in a stronger way, i.e., with more items per scale, allowing for more stable associations.

The relatively similar pattern of inter-relationships across all four pilots as depicted by the concurrent validities bolsters the interpretation that the scales function in relation to other scales in a relatively similar manner across countries. This implies that the meaning of the scales is also similar across the pilots in countries spanning the globe and thus attests to their utility as basic concepts in our still-developing vocabulary of psychosocial working condition effects.

The associations between JCQ 2.0 scales and health outcomes like burnout and depression are critical in understanding the impact of the work environment on employee wellbeing and have been empirically demonstrated in the JCQ 2 pilot studies [74,79]. This is significant for workplace interventions that focus on reducing occupational illness and promoting job variety, which—in turn—can help prevent disease and enhance wellbeing. By understanding the relationships between JCQ 2.0 scales and health outcomes, organizations can develop targeted interventions to promote a healthier work environment and support the wellbeing of their employees. This can ultimately lead to improved productivity, reduced absenteeism, and enhanced overall wellbeing.

One important interpretational issue arises in the context of the new organizational-level scales and is evident particularly in Australia and Germany: all organizational-level scales are rather highly correlated with each other, typically with values of about 0.45 (see also Formazin et al. [25]). One explanation for this, as noted in Karasek et al. [12], is that JCQ 2.0 respondents must integrate the vast multi-level detail of their general impressions of the organization’s complex structure, as seen from the “shop-floor-upwards,” into a unitary report, potentially leading to the high correlations across scales. However, we can also see that these correlations, while significant, also reveal much uncorrelated variance. Thus, the JCQ 2.0 organizational-level scales imply different pathways for both risk development and work environment change. From a work-redesign perspective, employee reports must certainly be supplemented with additional organizational level information to support effective work redesign.

### Strengths and Limitations

As a strength of our study, we note the application of the JCQ 2.0 in not only one or two but actually four countries. These four countries stem from three different continents, indicating our strive for covering very different work environments across the world.

The cross-sectional nature of data collection and, consequently, our concurrent validity associations represent a methodological limitation for causal interpretations of work-wellbeing effects. Moreover, response rates were comparably low in Australia and Germany. Nevertheless, the consistency of observed relationships, first at the item level, second at the recommended scale level, and third across countries—on the basis of comparisons of item and/or scale strength, of comparisons of risk and outcome associations, and of comparisons of differential strength of association with different health indicators and positive outcomes across cultures—altogether do support a pragmatically useful basis for multi-level workplace psychosocial assessment.

One limitation emerging from our cross-country study is that due to the stepwise development of the new JCQ 2.0 scales and items, the first pilot studies did not comprise all items, impeding combined analyses across the four countries. Furthermore, internal consistency was more often only moderate and sometimes weak for some of the scales in the early-stage pilots of Korea and China because not all items were available at that stage. Here, the tested final JCQ 2.0 scales were acceptable at the task level, but not at the organizational levels. This was not the case for the later pilots, where internal consistency was mainly “acceptable” to “good” when the scales were longer, indicating the costs and benefits associated with shorter vs. longer scales and the associated consequences for Cronbach’s Alpha. However, our approach led to clear scales by eliminating problematic items, which was a strength of our study.

A further JCQ 2.0 pilot design shortcoming was that dependent measures were not specified for the early pilots. Quite-general health outcomes were primarily used in Korea and China, limiting dependent variable associations with the more specific outcomes discussed in this paper that were based on more complete dependent variable scales from the later stage pilots. E.g., there were no positive outcome measures available in both countries. Hence, associations of the JCQ 2.0 scales with the work-related outcomes measures were weaker in the earlier pilots or could not be measured in Korea and China. Furthermore, Confucianism has traditionally guided the people in Asia to take the ‘middle way’ and they are less willing to select extreme responses, potentially reducing scale variance and associations [82,83,84]. Regarding the Korean pilot, low concurrent validity values may also have arisen because the sample was from one large municipal transit organization only and about 95% of participants were male, possibly leading to less variation among responses. This calls for further research in both countries. However, we feel it is unlikely that the JCQ 2.0 does not “function” in Korea or China. For example, the utility of the JCQ 1 has been demonstrated in Japan [2], in Taiwan [5], and in Thailand [85]. Our international comparison’s implicit prioritization of scale content over scale internal consistency brings the advantage of assessing similar content in both the independent and the dependent variables across very diverse cultural settings, but it yielded a limited question set for testing in the early pilots. Because the better, new, later-piloted JCQ 2.0 items were not available in the early pilots in Korea and China and scales were shorter, this led to lower internal consistency for the Korean and Chinese scale versions.

In the JCQ 2.0, employees assess work organization characteristics in terms of effects on their perceptions of the “work situation” as in the JCQ 1. While this can be considered a limitation due to potential self-report bias as, e.g., described by Karasek et al. [12] and Lazarus & Folkman [86], long-term use of the JCQ instrument in research has demonstrated the utility and reproducibility of this assessment. Nevertheless, it would be worthwhile to overcome this limitation in future research by applying triangulation techniques that also consider objective assessments of the work environment.

The JCQ 2.0 is based on the presumption that respondents can assess their particular “piece of work” in the organization as it appears to function from their perspective (i.e., on the shop floor). The JCQ 2.0 Respondent Instructions for the instrument’s organization-level scales state that the “organization” is referred to as “that aspect of the workplace where the rules are made about how you do your work”. Overall, JCQ 2.0 organization-level coverage is represented by a comprehensive collection of both in-the-literature and new ADC theoretically anchored scales, based on the DCS-S underlying structure as is tested in further empirical papers in this Special Issue [25]. The validity of these higher order self-report items must be tested by further cognitive interviews—a method to gain insight into how participants understand items—in the future.

## 5. Conclusions

The inclusion of cultural, social, and language translation in the psychometric scale development process has made the JCQ 2.0 scales robust and additionally given them the potential to capture relevant variations across different countries and cultures. Most instruments are initially developed in one language and tested in one culture and are unlikely to reflect, with sufficient precision, the cultural diversity and the translation issues that come with the performance of an instrument in terms of cross-cultural reliability and psychometric validity across countries. In contrast, the research and methodological design of the JCQ 2.0 involved translation and testing in four very different countries in similar ways (as depicted in Figure 1).

In conclusion, we find international support for comparable content and functionality across countries, which is further discussed in Formazin et al. [79]. Additionally, our results generally support the internal consistency and concurrent validity of the JCQ 2.0, especially in the Western culture of Australia and Germany. Further research should put its focus on replicating our results in other countries, applying the full set of JCQ 2.0 scales (which was not possible in our first pilots). Moreover, the research designs should allow for a longitudinal data collection, allowing for the causal interpretation of the data.

The substantially updated JCQ 2.0 task-level scales do appear to operate successfully across all four international pilots, including Korea and western China (albeit preliminarily)—a finding that is also seen in multiple country studies of the underlying JCQ 1. In the samples of Australia and Germany, the further extensions at the organizational level and even the external level, based on the extended theory in the ADC model and fully tested in the German version, do indeed demonstrate that the underlying extended ADC theory does support the new measurement structures at these higher levels. In Asian countries, the instrument’s higher-level scales require further testing in large representative samples.

Thus, the pilot tests provide both empirical support for the proposed multi-level JCQ 2.0 scale structure and empirical support for the ADC theory that underlies it. Altogether, across a broad international context, the new JCQ 2.0 can potentially support a substantial advance in the JCQ research tradition, better equipping researchers to study workplace factors in the modern free-market global economy.

## Figures and Tables

**Figure 1 ijerph-22-01435-f001:**
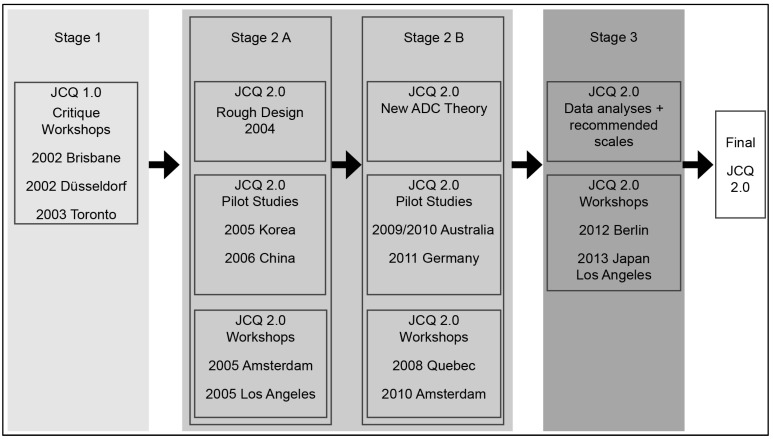
JCQ 2.0 developmental process.

**Figure 2 ijerph-22-01435-f002:**
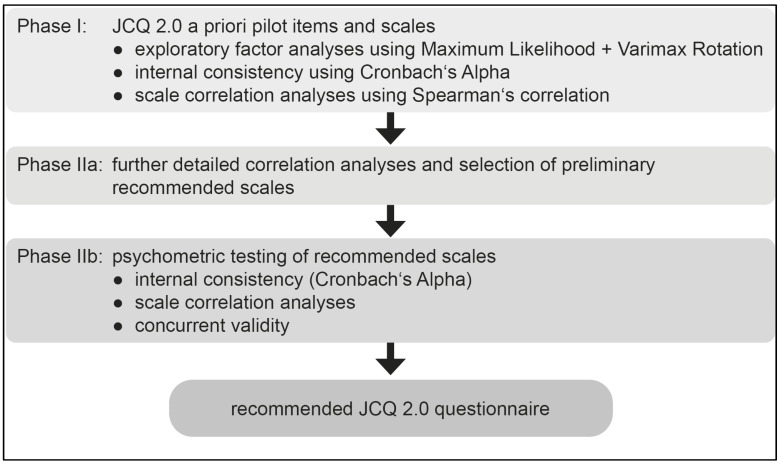
The process of validating the JCQ 2.0 questionnaire.

**Table 1 ijerph-22-01435-t001:** Overview of JCQ 2.0 scales.

	Demand	Control	Stability–Support
task level	quantitative demands (*r*)	skill discretion (*r*)	supervisor support (*r*)
	emotional demands (*l*)	decision authority	coworker support (*r*)
	physical demands (*r*)	conducive development (*n*)	collective control (*l*)
			negative acts (*n*)
organizational level	organizational restructuring (*n*)	organizational decision latitude (*n*)	psychosocial safety climate (*l*)
	organizational disorder (*n*)	procedural justice (*l*)	organizational rewards (*l*)
		conducive communication (*n*)	consideration of workers’ interests (*n*)
external-to-work level	job insecurity—personal (*r*)		
	job insecurity—career (*n*)		
	job insecurity—social relations (*n*)		
	global economy–demands and insecurity (*n*)		
	labor market control (*n*)		
	work–family conflict (*l*)		

*r* = Revision of JCQ 1 scale; *l* = new literature-based scale; *n* = new original JCQ 2.0 scale.

**Table 2 ijerph-22-01435-t002:** Demographics of participants in the four pilot studies.

Characteristics	Korea (*n* = 7407)		China (*n* = 2178)		Australia (*n* = 4214)		Germany (*n* = 2326)	
	Mean	SD	Mean	SD	Mean	SD	Mean	SD
Age (years)	42.1	3.9	42.8	6.8	45.7	12.2	45.4	10.4
Years work *	17.6	6.9	20.9	7.0	8.7	9.2	15.0	11.6
	Percentage (%)							
Gender								
Men	94.5		50.0		45.7		44.1	
Women	4.7		50.0		54.3		55.4	
Education level								
College and over	51.7		37.6		62.2		#	
Marital status								
Married/partnership	88.6		93.6		69.0		76.3	
Working sector								
Public	100		53.4		35.2		-	
Other	0		34.2		64.8		-	
Hours per week								
Full-time	100		64.3		59.6		72.1	
Part-time	0		23.7		23.3		27.4	

* in years of work = experience at current job. # = 81.0 with inclusion of vocational or technical certificate/diploma from a college and above

**Table 3 ijerph-22-01435-t003:** Cronbach’s Alpha coefficients for JCQ 2.0 scales.

Scale	No. of Items	Cronbach’s Alpha Coefficients			
		Korea	China	Australia	Germany
Task level					
Skill Discretion	3	0.59	0.67	0.64	0.70
Conducive Development	3 *	—	—	—	0.75
Decision Authority	3	0.66	0.53	0.71	0.76
Quantitative Demands	5	0.60 ^1^	0.40	0.60 ^2^	0.72
Emotional Demands	3	0.59	0.47 ^3^	0.74	0.57 ^4^
Physical Demands	5	0.67	0.67	0.79	0.87 ^5^
Supervisor Support	3	0.68 ^6^	0.62	0.84	0.85
Coworker Support	3	0.57 ^7^	0.79	0.85	0.82
Collective Control	3	— ^8^	0.26 ^9^	—	0.74
Negative Acts	2 *	—	—	—	0.74
Organizational level					
Conducive Communication	3 *	—	—	—	0.69
Organizational Decision Latitude	3	— ^10^	0.64	0.63	0.70
Procedural Justice	3	0.79	—	0.77	0.83
Organizational Rewards	2	—	—	0.50 ^11^	0.62
Consideration of Workers’ Interests	2 *	—	—	—	0.81
Psychosocial Safety Climate	4	—	—	0.86	0.90
Organizational Restructuring	3	—	0.30 ^12^	0.63	—
Organizational Disorder	4 *	—	—	—	0.71
External-to-work level					
Job Insecurity—Personal	3	0.31	0.74	—	0.79
Job Insecurity—Career	3	0.42	0.49	—	0.63
Job Insecurity—Social Relations	3 *	—	—	—	0.77
Work–Family Conflict	3	—	—	0.80	—
Global Economy—Demands and Insecurity	2 *	—	—	—	0.55
Labor Market Control	2 *	—	—	—	0.33

*: scale only available in Germany. —: not applicable. ^1^ *Free from work conflict* instead of *work conflict* available. ^2^ *easy* not available. ^3^ *acting* not available. ^4^ *acting* not available. ^5^ only country with *dangerous substances* and *accidents* available. ^6^ *respectful* not available. ^7^ *respectful* not available. ^8^ only *coworker unity* available. ^9^ *competition* not available. ^10^ only *influence* available. ^11^ *Respect and prestige* recoded is the same as *appreciation adequate*. ^12^ *cost cutting* not included because of 33% missing data.

**Table 4 ijerph-22-01435-t004:** Korea: Spearman’s correlations of JCQ 2.0 scales.

Scale	1	2	3	4	5	6	7
Task Level							
1. Decision Latitude							
2. Skill Discretion	0.74						
3. Decision Authority	0.81	0.20					
4. Quantitative Demands	0.04	0.05	0.09				
5. Emotional Demands	−0.13	−0.04	−0.15	0.06			
6. Physical Demands	−0.14	−0.10	−0.12	0.03	0.07		
Organizational level							
7. Organizational Decision Latitude	0.40	0.31	0.32	0.01	−0.06	−0.04	
8. Procedural Justice	0.27	0.17	0.24	0.13	−0.07	*−0.02*	0.27

All *p* < 0.01 unless noted by *italics.*

**Table 5 ijerph-22-01435-t005:** China: Spearman’s correlations of JCQ 2.0 scales.

Scale	1	2	3	4	5	6	7	8	9
Task Level									
1. Decision Latitude									
2. Skill Discretion	0.74								
3. Decision Authority	0.84	0.27							
4. Quantitative Demands	0.21	0.19	0.23						
5. Emotional Demands	0.06	0.08	*0.* *03*	0.32					
6. Physical Demands	−0.19	−0.21	−0.10	0.15	0.16				
7. Supervisor Support	0.26	0.19	0.22	0.23	*0.00*	−0.07			
8. Coworker Support	0.20	0.14	0.17	0.31	0.05	*−0.03*	0.39		
Organizational level									
9. Organizational Decision Latitude	0.37	0.23	0.34	0.25	0.06	−0.07	0.41	0.27	
10. Organizational Disorder	*0.* *02*	*0.02*	*0.* *01*	0.17	0.12	0.19	0.14	0.12	0.20

All *p* < 0.01 unless noted by *italics.*

**Table 6 ijerph-22-01435-t006:** Australia: Spearman’s correlations of JCQ 2.0 scales.

Scale	1	2	3	4	5	6	7	8	9	10	11	12	13
Task Level													
1. Decision Latitude													
2. Skill Discretion	0.81												
3. Decision Authority	0.86	0.42											
4. Quantitative Demands	0.04	0.14	−0.06										
5. Emotional Demands	*0.* *03*	0.13	−0.07	0.46									
6. Physical Demands	−0.23	−0.22	−0.18	0.12	0.11								
7. Supervisor Support	0.36	0.21	0.37	−0.22	−0.19	−0.18							
8. Coworker Support	0.30	0.22	0.29	−0.05	−0.05	−0.18	0.46						
Organizational level													
9. Organizational Decision Latitude	0.46	0.25	0.50	−0.17	−0.17	−0.11	0.43	0.21					
10. Procedural Justice	0.30	0.16	0.33	−0.25	−0.19	−0.09	0.45	0.23	0.57				
11. Organizational Rewards	0.26	0.13	0.30	−0.30	−0.27	−0.20	0.42	0.20	0.30	0.43			
12. Psychosocial Safety Climate	0.28	0.16	0.31	−0.29	−0.25	−0.11	0.52	0.22	0.54	0.62	0.41		
13. Organizational Restructuring	−0.05	0.04	−0.11	0.37	0.33	0.13	−0.20	−0.09	−0.24	−0.23	−0.25	−0.28	
External-to-work level													
14. Work–Family Conflict	0.04	0.13	−0.05	0.44	0.44	0.14	−0.17	−0.06	−0.12	−0.21	−0.24	−0.27	0.32

All *p* < 0.01 unless noted by *italics.*

**Table 7 ijerph-22-01435-t007:** Germany: Spearman’s correlations of JCQ 2.0 scales.

Scale	1	2	3	4	5	6	7	8	9	10	11	12	13	14	15	16	17	18	19	20	21	22	23
Task Level																							
1. Decision Latitude																							
2. Skill Discretion	0.78																						
3. Conducive Development	0.67	0.52																					
4. Decision Authority	0.88	0.39	0.60																				
5. Quantitative Demands	0.06	0.20	−0.11	−0.06																			
6. Emotional Demands	*0.02*	0.15	*−0.03*	−0.08	0.41																		
7. Physical Demands	−0.25	−0.19	−0.17	−0.22	0.23	0.17																	
8. Supervisor Support	0.30	0.13	0.42	0.34	−0.21	−0.19	−0.21																
9. Coworker Support	0.27	0.12	0.34	0.31	−0.10	−0.15	−0.11	0.47															
10. Collective Control	0.29	0.12	0.37	0.34	−0.22	−0.21	−0.22	0.45	0.62														
11. Negative Acts	−0.24	−0.08	−0.30	−0.30	0.16	0.26	0.25	−0.38	−0.44	−0.46													
Organizational level																							
12. Conducive Communication	0.45	0.29	0.58	0.44	−0.12	−0.10	−0.17	0.50	0.44	0.43	−0.31												
13. Organizational Decision Latitude	0.50	0.23	0.50	0.56	−0.12	−0.12	−0.14	0.50	0.38	0.44	−0.31	*0.51*											
14. Procedural Justice	0.40	0.20	0.45	0.43	−0.15	−0.11	−0.18	0.52	0.35	0.42	−0.27	*0.51*	*0.71*										
15. Organizational Fairness *	0.29	0.11	0.46	0.35	−0.35	−0.28	−0.25	0.59	0.39	0.44	−0.34	*0.54*	*0.52*	*0.53*									
16. Psychosocial Safety Climate	0.24	0.08	0.37	0.29	−0.27	−0.19	−0.17	0.48	0.30	0.34	−0.22	*0.43*	*0.44*	*0.45*	*0.62*								
17. Organizational Restructuring	−0.27	−0.13	−0.28	−0.30	*0.05*	0.08	0.18	−0.21	−0.22	−0.26	−0.21	*−0.29*	*−0.39*	*−0.30*	*−0.34*	*−0.23*							
18. Organizational Disorder	−0.30	−0.08	−0.40	−0.37	0.24	0.26	0.21	−0.48	−0.39	−0.46	0.35	*−0.47*	*−0.49*	*−0.51*	*−0.59*	*−0.45*	*0.35*						
External-to-work level																							
19. Job Insecurity—Personal	−0.20	−0.09	−0.24	−0.23	0.09	0.05	0.21	−0.24	−0.21	−0.30	0.25	*−0.21*	*−0.25*	*−0.23*	*−0.25*	*−0.21*	*0.20*	*0.22*					
20. Job Insecurity—Career	−0.20	−0.32	−0.60	−0.44	0.08	0.10	0.18	−0.45	−0.37	−0.45	0.35	*−0.51*	*−0.48*	*−0.45*	*−0.52*	*−0.40*	*0.24*	*0.42*	*0.44*				
21. Job Insecurity—Social Relations	*−0.03*	*0.02*	*−0.03*	−0.06	0.10	0.08	0.13	−0.10	−0.13	−0.19	0.19	*−0.01*	*−0.07*	*−0.05*	*−0.07*	*−0.03*	*0.05*	*0.10*	*0.41*	*0.16*			
22. Global Economy—Demands and Insecurity	−0.06	0.*02*	−0.13	−0.10	0.22	0.*05*	0.15	−0.15	−0.12	−0.26	0.13	*−0.12*	*−0.17*	*−0.18*	*−0.22*	*−0.19*	*0.09*	*0.14*	*0.44*	*0.26*	*0.25*		
23. Labor Market Control	0.31	0.14	0.36	0.35	−0.19	−0.12	−0.25	0.30	0.26	0.37	−0.26	*0.35*	*0.36*	*0.34*	*0.38*	*0.27*	*−0.24*	*−0.36*	*−0.43*	*−0.46*	*−0.56*	*−0.23*	
24. Work–Family Conflict	−0.19	0.81	−0.22	−0.11	0.41	0.29	0.16	−0.24	−0.19	−0.28	0.26	−0.17	−0.19	−0.21	*−0.27*	−0.24	0.61	0.32	0.17	0.89	*0.26*	0.21	−0.15

All *p* < 0.01 unless noted by *italics.* * Combination of the two scales’ organizational rewards and consideration of workers’ interests.

**Table 8 ijerph-22-01435-t008:** Concurrent validity coefficients (standardized odds ratio) with negative health outcomes for the JCQ 2.0 scales.

		Korea			China			Australia			Germany		
		Burnout	Depression	Poor SRH	Burnout	Depression	Poor SRH	Burnout	Depression	Poor SRH	Burnout	Depression	Poor SRH
Task level													
Skill Discretion		1.08	*1.00*	*1.04*	1.65	1.22	1.19	1.29	1.15	1.16	1.12	1.34	1.25
Conducive Development		—	—	—	—	—	—	—	—	—	2.05	2.17	1.78
Decision Authority		1.34	1.39	1.25	1.59	*1.06*	1.30	1.63	1.44	1.33	1.65	1.86	1.69
Quantitative Demands		1.11	1.16	*1.02*	1.17	*1.17*	*1.06*	2.27 #	1.66 #	1.31 #	2.68	1.59	1.53
Emotional Demands		1.59	1.39	1.40	1.36 #	1.23 #	*1.17 #*	2.19	1.80	1.44	2.03 #	1.53 #	1.49 #
Physical Demands		1.46 #	1.25 #	*1.46 #*	*2.06 #*	*1.36 #*	1.22 #	1.58 #	1.20 #	1.14 #	1.57	1.44	1.55
Supervisor Support		—	—	—	1.79	1.26	1.17	1.93	1.51	1.26	2.11	1.82	1.69
Coworker Support		—	—	—	1.55	*1.12*	1.18	1.43	1.38	1.20	1.58	1.55	1.48
Collective Control		—	—	—	—	—	—	—	—	—	2.04	2.00	1.85
Negative Acts		—	—	—	—	—	—	—	—	—	1.75	1.74	1.51
Organizational level													
Conducive Communication		—	—	—	—	—	—	—	—	—	1.87	1.97	1.69
Organizational Decision Latitude		1.17 #	1.18 #	1.11 #	1.51	*1.16*	1.27	1.86	1.54	1.35	1.93	1.95	1.72
Procedural Justice		1.37	1.39	1.17	—	—	—	1.96	1.54	1.32	1.90	1.82	1.60
Organizational Fairness *		—	—	—	—	—	—	2.12 #	1.78 #	1.44 #	3.10	2.41	2.21
Psychosocial Safety Climate		—	—	—	—	—	—	2.24	1.63	1.33	2.66	1.96	1.85
Organizational Restructuring		—	—	—	*1.09*	*1.03*	*1.14*	1.70	1.40	1.25	1.62 #	1.60 #	1.38 #
Organizational Disorder		—	—	—	—	—	—	—	—	—	2.36	2.11	1.73
External-to-work level													
Job Insecurity—Personal		—	—	—	—	—	—	—	—	—	1.56	1.67	1.55
Job Insecurity—Career		—	—	—	—	—	—	—	—	—	2.17	2.46	1.97
Job Insecurity—Social Relations		—	—	—	—	—	—	—	—	—	1.34	1.31	1.29
Work–Family Conflict		—	—	—	—	—	—	2.78	1.67	1.41	—	—	—
Global Economy—Demands and Insecurity		—	—	—	—	—	—	—	—	—	1.53	1.48	1.40
Labor Market Control		—	—	—	—	—	—	—	—	—	1.94	2.01	1.84

All *p* < 0.01 unless noted by *italics*; SRH—Self-rated Health; “—,” not applicable; #—Coefficients of scales with incomplete variables. Cases are considered as the approximately uppermost 20% of the scale. * Combination of the two scales organizational rewards and consideration of workers’ interests.

**Table 9 ijerph-22-01435-t009:** Concurrent validity coefficients (standardized odds ratio) with positive work outcomes for the JCQ 2.0 scales.

	Australia			Germany		
	Job Satisfaction	Job Engagement	High Intention to Stay	Job Satisfaction	Affective Commitment	Low Intention to Leave
Task level						
Skill Discretion	1.31	1.29	1.41	1.36	*1.13*	1.12
Conducive Development	—	—	—	3.40	2.08	1.89
Decision Authority	1.67	1.61	1.54	2.37	1.73	1.56
Quantitative Demands	1.58 #	1.29 #	1.32 #	1.34	1.13	1.67
Emotional Demands	1.46	1.24	1.35	1.42 #	*1.06 #*	1.54 #
Physical Demands	1.16 #	*1.04 #*	1.13 #	1.78	*1.12*	1.34
Supervisor Support	2.05	1.57	1.58	3.78	2.26	2.33
Coworker Support	1.64	1.42	1.35	2.51	1.51	1.85
Collective Control	—	—	—	2.88	1.48	2.08
Negative Acts	—	—	—	2.83	1.41	1.80
Organizational level						
Conducive Communication	—	—	—	3.84	2.18	1.94
Organizational Decision Latitude	1.81	1.71	1.68	2.91	2.02	1.96
Procedural Justice	2.15	1.72	1.68	2.82	1.84	1.99
Organizational Fairness *	2.19 #	1.59 #	1.71 #	4.58	2.41	2.94
Psychosocial Safety Climate	2.39	1.73	1.70	2.59	1.96	2.39
Organizational Restructuring	1.69	1.35	1.39	2.00 #	1.56 #	1.26 #
Organizational Disorder	—	—	—	3.74	2.21	2.33
External-to-work level						
Job Insecurity—Personal	—	—	—	2.09	1.57	2.12
Job Insecurity—Career	—	—	—	4.90	2.13	2.91
Job Insecurity—Social Relations	—	—	—	1.39	1.14	1.45
Work–Family Conflict	1.64	1.32	1.29	—	—	—
Global Economy—Demands and Insecurity	—	—	—	1.47	*1.12*	1.62
Labor Market Control	—	—	—	2.52	1.53	1.86

All *p* < 0.01 unless noted by *italics*; “—” not applicable; #—Coefficients of scales with incomplete variables. Cases are considered as the approximately uppermost 20% of the scale. * Combination of the two scales organizational rewards and consideration of workers’ interests.

## Data Availability

Due to data privacy regulations, the data from Germany, Korea, and China cannot be distributed. Australian Workplace Barometer Data is available at The Australian Data Archive, Australian National University.

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
