# Peer review of "An International Comparative Reliability and Concurrent Validity Assessment of the Multi-Level Job Content Questionnaire (JCQ) 2.0"

_ijerph, 2025, doi:10.3390/ijerph22091435_

Round 1
Reviewer 1 Report
Comments and Suggestions for Authors
Thank you for inviting me to review this work titled "International Comparative Reliability and Concurrent Validity Assessment of the Multi-level Job Content Questionnaire (JCQ) 2.0". Although this paper would belong to a special issue that includes complementary papers, and furthermore reports methodology and results of 4 pilots, I present some major concerns that warrant attention before publication:
- In the abstract lines 27-28 it states that the objective of this study was "to facilitate understanding of cross-cultural disparities". I am not sure that this paper can facilitate such an understanding because 1) sample composition varies considerably (e.g., Korea almost only men, all from one public institution); 2) scale variety (given the evolutionary process of the JCQ 2.0 development, MOST of the new scales cannot be compared across pilots/countries); 3) the methodology (Despite reporting Cronbach's alpha, Spearman’s correlations and SOR results (although appropriate), not much insights for "understanding cross-cultural disparities" are reported. Factorial validity is mentioned but outside the scope of this paper, more rigorous testing of the adaptation and psychometric cross-cultural validation of the scales, for example by multi-group CFAs, testing levels of invariance, or latent means differences, is missing.). Theorizing on cultural differences is limited. It seems that the authors suggest general utilitly of the JCQ 2.0, (l. 599), yet more empirical evidence for this claim is warranted.
- The most recent pilot had been conducted in 2011, in an arguably significantly different economic, technological, and social context (post-2008 crisis, job instability, etc.). I wonder if the JCQ 2.0 accounts well for today's context that is as diverse as probably ever, with certain employee segments suffering from underemployment (precarious conditions), others from overemployment (overload), pluri-employment, increasing nonwork demands (informal caregiving duties), "always-on" cultures and prolonged psychophysiological activation (undermined recovery processes; blurred boundaries), and ever more unstable political, natural, and social environments. Although the 25 scales seem to validly measure important psychosocial risk factors and potentially impactful resources, I am not convinced that these 25 are the most important given the vast variety of individual work stress situations laid out above.
- Although the ADC model expands the DC model and (arguably) adds to existing open systems theories, the general multilevel theory is not really tested by the use of multilevel modeling that would account for the nested data context. If, in the end, the JCQ2.0 is applied as a self-report tool, I assume this implies talking about "perceptions" and represents second-order cybernetic thinking. Given the latter, this would rather support subjective appraisal theories. Thus, I believe it would be of utmost importance to align the methodology with the newest ADC theoretical conjectures. This may be beyond the scope of this paper.
Minor concerns:
- Quality of Figure 1 and 2, especially 1, should be improved.
- Abbreviation before explanation in l. 102
- Typos, for example in l. 73 and l. 365
- unfinished sentences, for example in l. 274
- wrong definition of engagement/commitment in l. 301
- confusing chunks like "are available at [online references] at___)[67]." in l. 384
- "predictive" should not be used in l. 413 and fig. 2
- I do not understand the sentence in lines 691-695
- Which "cognitive procedures" in l. 764 do you mean?
- "simultaneous" in l. 786 is misleading because data was collected at very different time points.
- what you refer to when mentioning "Western-culture advanced economies" in l. 791?
Thank you for this insightful work on how the JCQ2.0 has been investigated theoretically and empirically. Good luck.
Reviewer 2 Report
Comments and Suggestions for Authors
Thank you for the opportunity to review the manuscript titled "International Comparative Reliability and Concurrent Validity Assessment of the Multi-level Job Content Questionnaire (JCQ) 2.0." I commend the authors for undertaking an ambitious, multi-country psychometric study on a timely and significant topic—developing an instrument that captures the evolving nature of psychosocial work environments across task, organizational, and external domains. The inclusion of data from four diverse national contexts and the integration of a multi-level theoretical framework (including ADC theory) are notable strengths. The paper also contributes by expanding traditional psychosocial models to address contemporary stressors related to globalization and organizational change. I appreciate the opportunity to engage with this important work and provide constructive feedback to support its further development.
Title
Weaknesses:
-
Too long and overly technical.
-
Lacks clarity and accessibility for general readers.
-
Does not highlight the multi-level innovation or new theoretical basis (ADC theory).
-
Misses practical relevance (e.g., workplace health, cross-cultural application).
Abstract
Suggestions for Improvement:
-
Clearly divide into: Background, Methods, Results, and Conclusion.
-
Use plain, direct language (e.g., replace "psychometrically assessed" with "evaluated for reliability and validity").
-
Include at least one specific result (e.g., Cronbach’s alpha range, SOR findings).
-
End with a focused conclusion on the tool’s utility and next steps.
Introduction
Weaknesses:
-
Lack of focus and coherence (Major):
-
The introduction covers too much conceptual ground without sufficient focus. Multiple theories are introduced in rapid succession (e.g., Demand-Control, Stress-Disequilibrium, ADC, PSC, ERI) without clarifying their distinct contributions or how they interrelate.
-
It reads more like a theoretical literature dump than a logically organized rationale.
-
-
Missing gap articulation (Major):
-
It does not clearly identify the specific empirical or theoretical gap the JCQ 2.0 is addressing. For example: What specific limitations of JCQ 1.0 were observed in prior cross-national studies? What new constructs are missing in existing psychosocial work assessment tools?
-
-
Unclear research objectives (Minor):
-
While it mentions the use of psychometric analysis across four countries, it lacks specific research questions or hypotheses.
-
-
Redundancy and verbosity (Minor):
-
Phrases like “holistic understanding of stressors of the contemporary work environment improving occupational health promotion, work redesign, wellness programs, and organizational interventions” are overloaded and repetitive.
-
Methods
Weaknesses:
-
Overly complex and fragmented structure (Major):
-
The Methods section is overloaded with sub-subsections (e.g., 2.2.1, 2.4.2, 2.4.3) that interrupt the flow.
-
Important methodological logic (e.g., scale refinement criteria) is embedded deep in the text and should be summarized more clearly.
-
-
Lack of clarity in scale selection process (Major):
-
The five criteria for final item selection are buried and not summarized concisely. A table or flow diagram would help clarify the stepwise process across pilots.
-
The rationale for dropping or retaining specific items (e.g., “work hard,” “repetitive work”) is mentioned but not justified with psychometric data.
-
-
Insufficient explanation of cross-national comparability strategy (Minor):
-
While differences in item wording and scale coverage across countries are acknowledged, the paper does not detail how these variations were controlled for or how they affected comparability.
-
-
Inconsistent use of statistical terms (Minor):
-
Some terms (e.g., “concurrent validity,” “standardized odds ratios”) are used without sufficient introduction or definition for non-specialist readers.
-
Methods
Weaknesses:
-
Overly complex and fragmented structure (Major):
-
The Methods section is overloaded with sub-subsections (e.g., 2.2.1, 2.4.2, 2.4.3) that interrupt the flow.
-
Important methodological logic (e.g., scale refinement criteria) is embedded deep in the text and should be summarized more clearly.
-
-
Lack of clarity in scale selection process (Major):
-
The five criteria for final item selection are buried and not summarized concisely. A table or flow diagram would help clarify the stepwise process across pilots.
-
The rationale for dropping or retaining specific items (e.g., “work hard,” “repetitive work”) is mentioned but not justified with psychometric data.
-
-
Insufficient explanation of cross-national comparability strategy (Minor):
-
While differences in item wording and scale coverage across countries are acknowledged, the paper does not detail how these variations were controlled for or how they affected comparability.
-
-
Inconsistent use of statistical terms (Minor):
-
Some terms (e.g., “concurrent validity,” “standardized odds ratios”) are used without sufficient introduction or definition for non-specialist readers.
-
Results
Weaknesses:
-
Lack of synthesis and interpretation (Major):
-
The section reads like a data dump. It presents metrics but offers minimal interpretation or synthesis. For example, it notes that 73% of German scales have α > .70 but does not explain what that implies about instrument robustness.
-
There is no summary of key patterns across countries or levels (task, organizational, external).
-
-
Missing justification of thresholds (Minor):
-
While standardized odds ratios (SORs) are used, the justification for a 1.55 cut-off is buried and lacks citation strength. Readers unfamiliar with SORs need context.
-
-
Inconsistent emphasis on poor results (Minor):
-
Some weak results (e.g., low α in China and Korea) are downplayed, while stronger results (e.g., in Germany and Australia) are highlighted. A balanced discussion of strengths and limitations is needed.
-
-
Over-reliance on tables without narrative integration (Major):
-
The text refers readers to multiple tables (e.g., Table 8, Table 9) but does not walk them through the key findings. This forces the reader to interpret raw data independently, reducing accessibility.
-
Suggestions for Improvement:
-
Begin the section with a brief overview of the strongest and weakest performing scales across countries.
-
Explicitly discuss how JCQ 2.0 outperforms JCQ 1.0 in robustness or scope (if supported).
-
Highlight cross-cultural consistencies and discrepancies with interpretation (e.g., why Chinese results were weaker).
-
Consider a summary figure or matrix comparing reliability and validity scores across countries and levels.
Discussion
Weaknesses:
-
Overly conceptual and under-empirical (Major):
-
The discussion disproportionately emphasizes theory (ADC, Conducivity, PSC, etc.) rather than interpreting the actual empirical findings.
-
It lacks integration of key results: for example, what does the high internal consistency in Germany and weak performance in Korea imply for generalizability?
-
-
Insufficient engagement with limitations (Major):
-
Some known weaknesses (e.g., variation in item availability across countries, low Cronbach’s alpha in some scales, low response rates in Australia and Germany) are not critically analyzed.
-
No reflection on the limitations of concurrent validity as a psychometric criterion.
-
-
Missed opportunity for comparison to existing tools (Minor):
-
No comparative positioning against other global work stress scales (e.g., COPSOQ, ERI short scales), which would enhance the perceived contribution of JCQ 2.0.
-
-
Weak conclusions and future directions (Minor):
-
The paper closes with generalized claims about sustainability and wellness but does not propose concrete future research or implementation strategies (e.g., longitudinal validation, sector-specific applications).
-
Conclusions
Weaknesses:
-
Overly general and repetitive (Major):
-
The conclusion restates earlier claims without distilling the most important insights from the results.
-
Phrases like “supports promotion of worker health and wellbeing” are vague and could apply to any psychosocial instrument.
-
-
No clear summary of contributions (Major):
-
There is no succinct wrap-up of what the study achieved that previous efforts did not.
-
It lacks a statement of how JCQ 2.0 improves on JCQ 1.0 in measurable ways (e.g., new scale performance, cross-cultural robustness).
-
-
No mention of limitations or caution (Minor):
-
A good conclusion should acknowledge at least one key limitation to avoid overclaiming validity. This is especially important for self-developed instruments tested in pilot contexts.
-
-
Missed opportunity to frame next steps (Minor):
-
There’s no mention of practical follow-up: no proposal for longitudinal testing, implementation in different sectors, or use in non-OECD countries.
-
In conclusion, this manuscript makes a valuable contribution to the field of occupational health by advancing the measurement of psychosocial work conditions through a theoretically rich and empirically grounded tool. However, the paper would benefit from significant revisions to improve clarity, focus, and accessibility—particularly in the title, abstract, methods, results, and discussion sections. Addressing these issues will enhance the manuscript’s scientific rigor and practical relevance, especially for an international readership. I thank the authors once again for their efforts and the editor for the opportunity to review this important work. I look forward to seeing a revised version that fully realizes the potential of this ambitious study.
Comments on the Quality of English LanguageWhile generally grammatically correct, the writing is often dense, overly technical, and occasionally repetitive. Key ideas are sometimes obscured by jargon or long, complex sentences. Clearer structure and more concise phrasing would improve readability and accessibility for a broader academic audience.
Reviewer 3 Report
Comments and Suggestions for Authors
I found myself persuaded by your careful presentation of your thoughtfully designed study. You do seem to have advanced this job content questionnaire/ instrument in a thoughtful way. That said, and while I realize it is not your central focus, I think it would be helpful to your readers to provide some specific examples in your introduction of the ways in which the broader socio-economic context has been evolving that make this effort significant. It also seems important to offer some specific examples of how organizations and public authorities can and should employ information from these studies to support their citizens. I also found myself curious concerning what sorts of "additional studies" you would advocate to validate more robustly this update and where and why.
Comments on the Quality of English LanguageThe article is generally quite legible if replete with relatively arcane methodological discussion (which is unavoidable), but there are numerous typographic errors, several sentence fragments and one or two incorrect word choices. Examples may be found at lines 73, 120, 273, 367, 559,635 and 661 for example. The piece needs a careful line edit throughout.
Round 2
Reviewer 2 Report
Comments and Suggestions for Authors
The authors have successfully revised the manuscript.